# Odor classification: Exploring feature performance and imbalanced data learning techniques

**Durgesh Ameta**[1,2‡*], **Surendra Kumar**[3‡], **Rishav Mishra**[3‡], **Laxmidhar Behera**[1,4‡], **Aniruddha Chakraborty**[5‡], **Tushar Sandhan**[4‡]

**1** Indian Knowledge System and Mental Health Applications Centre, Indian Institute of Technology, Mandi, India, **2** Indian Knowledge System Centre, ISS, Delhi, India, **3** School of Electronics, Indian Institute of Information Technology, Una, India, **4** Department of Electrical Engineering, Indian Institute of Technology, Kanpur, India, **5** School of Basic Sciences, Indian Institute of Technology, Mandi, India

‡ DA contributed to Conceptualization, Data collection, Analysis, and paper writing. SK and RM contributed to Data collection, Analysis, and paper writing. LB and AC contributed to Supervision, Interpretation, Review and editing. TS contributed to Conceptualization, Supervision, Interpretation, Review and editing.
* durgesgameta@gmail.com

**Data availability statement:** The fingerprint data (IGD_FP and Subset-IGD_FP) used in this study is a compiled dataset created by merging two separate expertly labelled odor datasets. The Chemistry WebBook, sponsored by the

## Abstract

This research delves into olfaction, a sensory modality that remains complex and inadequately understood. We aim to fill in two gaps in recent studies that attempted to use machine learning and deep learning approaches to predict human smell perception. The first one is that molecules are usually represented with molecular fingerprints, mass spectra, and vibrational spectra; however, the influence of the selected representation method on predictive performance is inadequately documented in direct comparative studies. To fill this gap, we assembled a large novel dataset of 2606 molecules with three kinds of features: mass spectra (MS), vibrational spectra (VS) and molecular fingerprint features (FP). We evaluated their performance using four different multi-label classification models. The second objective is to address an inherent challenge in odor classification multi-label datasets (MLD)—the issue of class imbalance by random resampling techniques and an explainable, cost-sensitive multilayer perceptron model (CSMLP). Experimental results suggest significantly better performance of the molecular fingerprint-based features compared with mass and vibrational spectra with the micro-averaged F1 evaluation metric. The proposed resampling techniques and cost-sensitive model outperform the results of previous studies. We also report the predictive performance of multimodal features obtained by fusing the three mentioned features. This comprehensive and systematic study compares the predictive performance for odor classification of different features and utilises a multifaceted approach to deal with data imbalance. Our explainable model sheds further light on features and odour relations. The results hold the potential to guide the development of the electric nose and our dataset will be made publicly available.

National Institute of Standards and Technology (NIST), provided the mass spectra (Subset-IGD_MS) and vibrational spectra (Subset-IGD_VS) used in this investigation. Supplementary File 1 contains a list of the molecules in the IGD and Subset-IGD categories, and Supplementary File 2 has the specifics of how the categorization models were implemented.

**Funding:** Ministry of Education at AICTE, IKS Centre project (2-28/IKS Center-2/2022-23/54). Tata Consultancy Services, TCS Research Scholar Grant.

**Competing interests:** The authors declare no competing interests.

# 1 Introduction

Smell and taste, two fundamental senses among the five, play an essential role in detecting and recognizing chemicals. Scientists and researchers are intrigued by the complex nature of the sense of smell and its close relationship to memory and emotions; moreover, they have consistently endeavoured to investigate its underlying mechanisms. In the olfactory epithelium, which is located in a 3.7 cm$^2$ region in the upper nasal passageways, olfactory receptor neurons interact with volatile odorant molecules to help perceive odors. [1]. Linda Buck and Richard Axel's Nobel Prize-winning research uncovered olfactory receptors (ORs), which are members of the G-Protein Coupled Receptor class [2]. Higher brain regions process the patterns created by the aggregation and transport of signals from activated olfactory neuron circuits to the olfactory bulb. The significant advancements in speech, audition, and vision suggest that the sensory output for olfactory stimuli can be predicted similarly [3]. Recent years have witnessed an increasing interest in leveraging data-driven approaches, particularly in the realm of machine learning, for predicting structure-odor relationships [4–9]. Numerous endeavours have aimed to showcase the viability of predicting olfactory perception using structural features, typically encoded by molecular fingerprints, mass spectra, and vibrational spectra of the odorant molecules [7–9]. Among these works, datasets, features and classification approaches differ significantly.

Dravnieks and colleagues created a database comprising 160 odorants, while Keller et al. developed the DREAM dataset, consisting of 480 structurally diverse molecules with continuous values assigned to sensory attributes such as odor intensity and pleasantness [10,11]. On the other hand, both Goodscents and Leffingwell datasets have more than three thousand molecules, and 113 unique odor descriptors, the sensory data as binary presence or absence of odor descriptor [12,13].

For featurisation, various properties of odorants are encoded differently. Fingerprint features mainly capture physical properties such as freezing point, boiling point, density, solvation properties, and molecular attributes [8]. According to the vibrational theory of olfaction, a molecule's infrared vibrational frequency and its olfactory characteristics are strongly correlated [7]. Hence, VS is also used as a feature for odor classification. The physicochemical features capture molecular structure parameters, but they cannot be used for chemical mixtures; mass spectrum can express the mixture; hence, it is also used as a feature [9]. Some commonly used fingerprint features in recent studies include physicochemical properties and SMILES representation. Tools like Mordred (open source) and Dragon (closed source) can generate thousands of physicochemical properties for each molecule, including details about functional groups, atom types, topological indices, and ring descriptors, among other molecular characteristics, and they are used for ML/DL analysis [8]. The cheminformatics program RDKit can also be used to calculate the Simplified Molecular-Input Line-Entry System (SMILES), another popular technique [14]. From a SMILES representation, we can derive molecular fingerprints, which come in two types: bit-based fingerprints, which are binary representations indicating the presence or absence of certain substructures in the molecule, and count-based Morgan fingerprints (CFP), which provide counts of particular substructures in the molecule, offering a more detailed representation. These methods and tools are crucial for analyzing and characterizing molecular structures in cheminformatics [8].

In the literature, various methods for analysing features have been employed [15]. Among these, Licon et al. introduced a database comprising 1689 odorants and proposed a computational method based on subgroup discovery algorithms to search rules connecting chemistry with perception [16]. Lee et al. employed the same Graph Neural Network (GNN) to develop a scent map, analogous to colour maps, positioning molecules with comparable scents in

proximity [17]. Their findings further illustrated the GNN's ability to generalize across new datasets and various scent-prediction tasks. Furthermore, Liu et al. presented DeepSniffer, a framework aimed at classifying scents in essential oils using a Multilayer Perceptron (MLP) that is based on k-shot meta-learning [18]. In order to create machine learning algorithms that can precisely predict sensory attributes, including odor intensity, pleasantness, and eight semantic descriptors for 480 structurally diverse odorant molecules, Keller and colleagues organized the crowdsourced DREAM Olfaction Prediction Challenge [10]. To predict patients' perceptual ratings based on 4,884 molecular parameters, they trained 1,089 regression models using a random forest approach. The correlation between the observed and projected scores served as the performance metric [10]. In parallel, Zhang and co-authors created predictive models based on Convolutional Neural Networks (CNNs) to analyze odor characteristics and pleasantness. Their objective was to develop a process for creating and evaluating fragrance molecules, each of which was to be represented by a distinct olfactory character [6].

From the literature survey, we find three broad classes of features used for odor prediction, mass spectra, as they can be used for both mono-molecular and chemical mixture, vibrational spectra not only capture different molecular features but also record vibrational frequency, which plays a role in transferring the vibrational energy, which, according to the vibrational theory of olfaction, plays an important role along with other structural properties of molecules, third is physicochemical features, which are frequently used in cheminformatics studies to encode the features of molecules. Hence, we use these three classes of features for analysis. We assembled a novel dataset with sensory data as the binary presence of odor descriptors used in this study and used three baseline multi-label models—Random Forest (RF) [19], Binary Relevance (BR) [20], Classifier Chain (CC) [21]—and a cost-sensitive MLP (CSMLP) for classification [22].

This paper's main contributions are as follows: (1) We compiled a large and novel dataset of having VS, MS and physicochemical features; (2) a systematic comparison of the predictive performance of different features for odor classification; (3) we used random resampling techniques to address data imbalance in the odor datasets; (4) exploratory analysis of multi-modal feature fusion for odor classification; (5) an explainable deep model provides insight into the feature odor relationship.

The paper is organized as follows: The datasets and feature creation process are described in the first section, which also describes the multi-label classification models employed for classification and the resampling methods. The following section describes Shapley Additive Explanations (SHAP) feature importance method, followed by the results of classification models, feature importance, and feature fusion analysis. The last section contains the conclusions.

## 2 Materials and methods

This section begins by detailing the datasets and their featurization processes. Subsequently, it elaborates on the classification methods and discusses various random resampling techniques.

### 2.1 Dataset

Characterizing odorants entails taking into account a number of qualitative descriptors, such as hedonic qualities like "fruity" and "fishy," which express how pleasant or disagreeable a smell is. We used a combined dataset comprised of the Leffingwell dataset and the training data supplied by Firmenich during the "Learning to Smell Challenge". We call this as Integrated dataset (IGD), which consists of 109 different odor classes and 7,374 molecules with

substantial structural variation. A Subset of the Integrated Dataset (Subset-IGD) comprising 2,606 odorant molecules was also produced by us. We obtained Daylight Fingerprints, Vibrational Spectra (VS), and Mass Spectra (MS) for this subset.

**Integrated dataset (IGD).** Two expertly labeled odor datasets were combined to create IGD by Saini et al. [8]. These two datasets are Leffingwell and the training materials Firmenich supplied for the "Learning to Smell Challenge". Following preprocessing and merging, the final dataset had 7,374 molecules spread across 109 different odor classes, each with a different sample count (Fig 1). Supplementary File 1 contains the full list of all 7,374 odorant compounds.

**Subset of Integrated Dataset (Subset-IGD).** Using SMILES code, we generated the International Chemical Identifier (InChI) Code and CAS Registry number for molecules in the Integrated Dataset (IGD). NistChemPy Python package was used to convert SMILES to CAS or InChI. Then, we searched for the Vibrational Spectra (VS) and Mass Spectra (MS) of every molecule in the IGD on the National Institute of Standards and Technology's (NIST) Chemistry WebBook. Despite IGD containing 7,374 molecules, only 2,606 unique molecules with MS and VS were identified, each associated with a total of 109 distinct odors, presenting a multi-label classification challenge. As a result, this subset of the Integrated Dataset, termed Subset-IGD, was formed to represent this smaller yet representative subset of IGD. The distribution of odor classes within both IGD and Subset-IGD is depicted in Fig 1, showing the proportion of samples associated with each odor class. Subset-IGD serves as a representative subdataset of IGD. This conclusion is supported by the Kolmogorov-Smirnov (K-S) test, which produces a p-value of 0.75 and a test statistic (D) of 0.0917. Supplementary file 1 contains a list of all 2606 odorant molecules.

## 2.2 Featurising molecules

We transformed molecules into meaningful numerical representations to train our classification models. Traditional Daylight fingerprinting was used for this featurization, as it provided the best results in previous studies with the dataset. Furthermore, vibrational and mass spectrum features were also obtained, the methodologies of which are discussed in the following sections.

**Subset-IGD_VS (Vibrational Spectra features).** The National Institute of Standards and Technology (NIST) hosts the Chemistry WebBook, which is the source of the VS matching to the derived CAS and InChI of molecules in IGD [23]. Following the optimization of the molecular structures to attain minimum energy, the vibrational frequencies and intensities of each of these molecules were calculated using the B3LYP/6-31G* level of theory, a form of density functional theory (DFT). For these DFT computations, the Gaussian 09 suite [24] was utilized. For each spectrum, a set of wavenumbers (vwns) related to a specific molecule is projected onto a bounded frequency scale (BFS), typically ranging from 1 to 4000 cm$^{-1}$. For further analysis, this range is chosen to include all necessary vibrational normal modes. The spectra are then subjected to Gaussian smoothing. To aid with dimensionality reduction, the BFS is normalized and sampled at predetermined increments of Lcm$^{-1}$ [25].The generated spectra—henceforth referred to as Subset-IGD_VS—are utilized as features for the classification models. The following is the formula for a Gaussian kernel that smoothes the VS:

$$K_{\text{gaussian}} = \frac{1}{\sigma\sqrt{2\pi}} \exp\left(\frac{(x-\bar{x})^2}{2\sigma^2}\right) \tag{1}$$

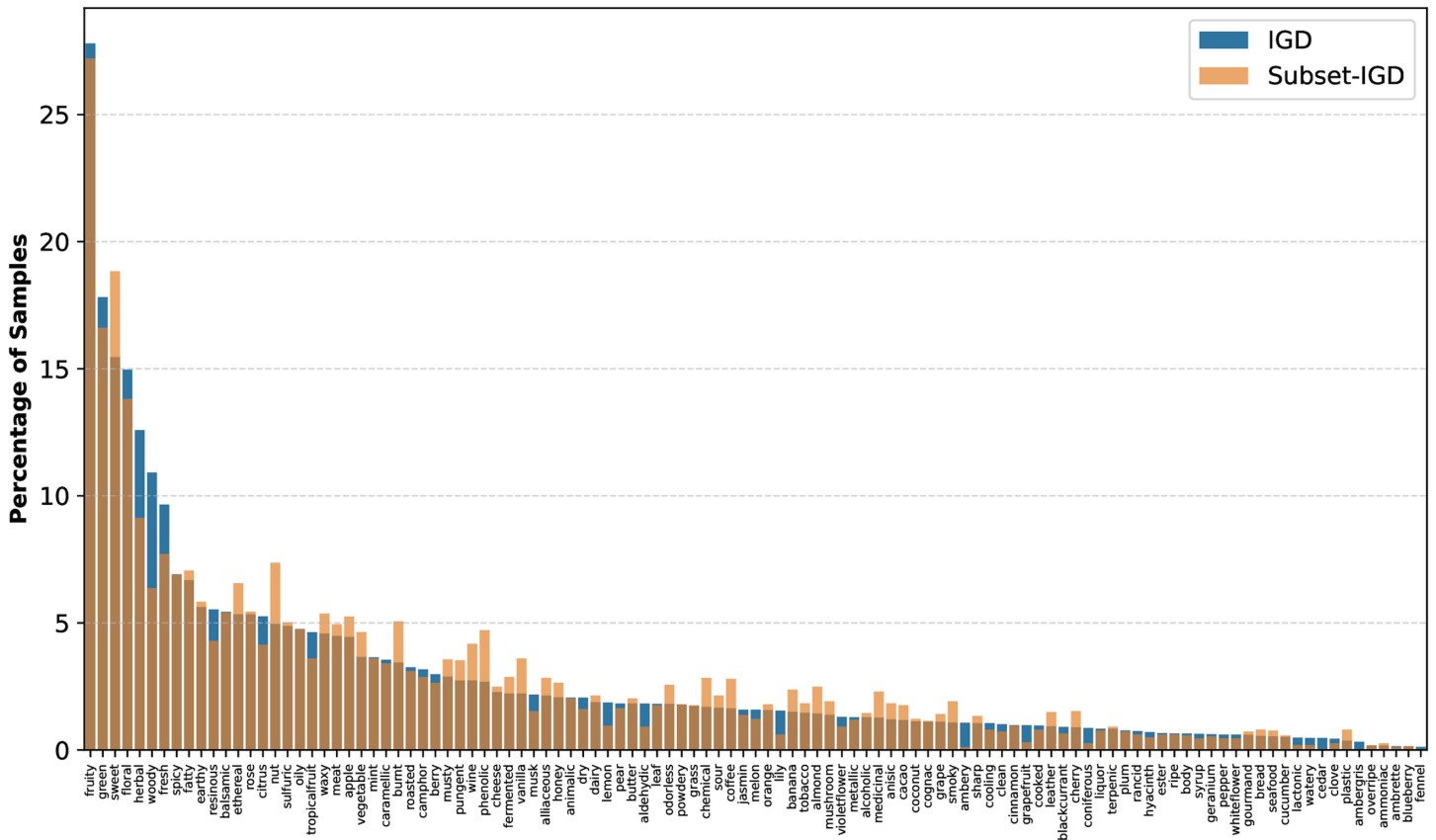

**Fig 1.** Plot shows the proportion of samples associated with each odor class in Integrated Dataset (IGD) and Subset of Integrated Dataset (Subset-IGD). There is a significant imbalance in associated samples of odor labels, with the "fruity" label appearing around 27% of the time, whereas the occurrence of "fennel" is less than 1% in the dataset.

A sigma value of 10 cm$^{-1}$ and an L value of 5 cm$^{-1}$ were used, resulting in 800 descriptor variables (calculated as 4000/L). The complete process of extracting features from VS is illustrated in Figure 2. The smoothing procedure utilizes a smearing function [26] that broadens the vibrational peaks, enabling the comparison of different compounds with slightly varying frequencies. This featurization technique for VS spectra was introduced by Turner et al. and has been validated for use in Quantitative Structure-Activity Relationship (QSAR) studies [26]. Hence, Subset-IGD_VS finally had 2606 molecules with 800 features each [25].

**Subset-IGD_MS (Mass Spectra features).** Mass spectra (MS) corresponding to the obtained CAS and InChI identifiers were retrieved from the Chemistry WebBook provided by the National Institute of Standards and Technology (NIST) [23]. The original mass spectrum from NIST consists of over 300 dimensions. However, our analysis focused specifically on the intensity range of 50–262 m/z (mass-to-charge ratio). This selection was made because intensity values below 50 m/z are primarily associated with odorless molecules like oxygen, while intensities above 262 m/z are linked to low-volatility molecules. Consequently, the resulting data matrix contains 2,606 samples (rows) and 213 intensity values (columns). This mass

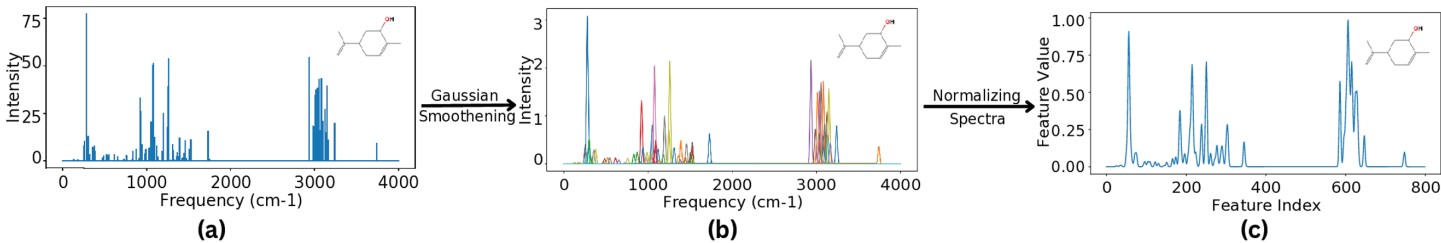

**Fig 2. Overview of feature generation from VS for molecule Carveol (shown in inset).** **(a)** projected VS onto a BFS, **(b)** spectra obtained after Gaussian smoothing, **(c)** final VS spectra with 800 features obtained after normalizing spectra with BFS sampled at $5cm^{-1}$. Obtained spectra in **(c)** are used as features for classification models.

spectra data matrix was normalized to a range of 0 to 1 by dividing each intensity value by the maximum intensity value within the corresponding mass spectrum. The complete process of feature extraction from the mass spectra is illustrated in Fig 3. The 213 features from the 2,606 molecules will henceforth be referred to as Subset-IGD_MS.

**Subset-IGD_FP (Finger-Print features).** The daylight fingerprint features were obtained utilizing the RDKit library to extract them from SMILES-encoded molecular structures. This involved converting the SMILES strings into molecular structures and generating fingerprints to capture key chemical substructures. These fingerprints were then converted into a binary format for efficient analysis. This approach enabled us to extract essential structural information from the molecular data, facilitating further analyses and modeling. Subset-IGD_FP denotes a dataset with 2,606 molecules with 1024 features of the Daylight fingerprint features.

**IGD_FP (Finger-Print features for IGD).** The procedure for acquiring IGD_FP is the same as that of Subset-IGD_FP, just applied to the entire Integrated Dataset. IGD_FP represents 1024 features for each molecule of IGD.

## Data analysis

A systematic analysis of Subset-IGD_VS, Subset-IGD_MS, Subset-IGD_FP, and IGD_FP classification with resampling was performed, and also saliency analysis of the model was done using SHapley Additive explanations [27] to explain model predictions.

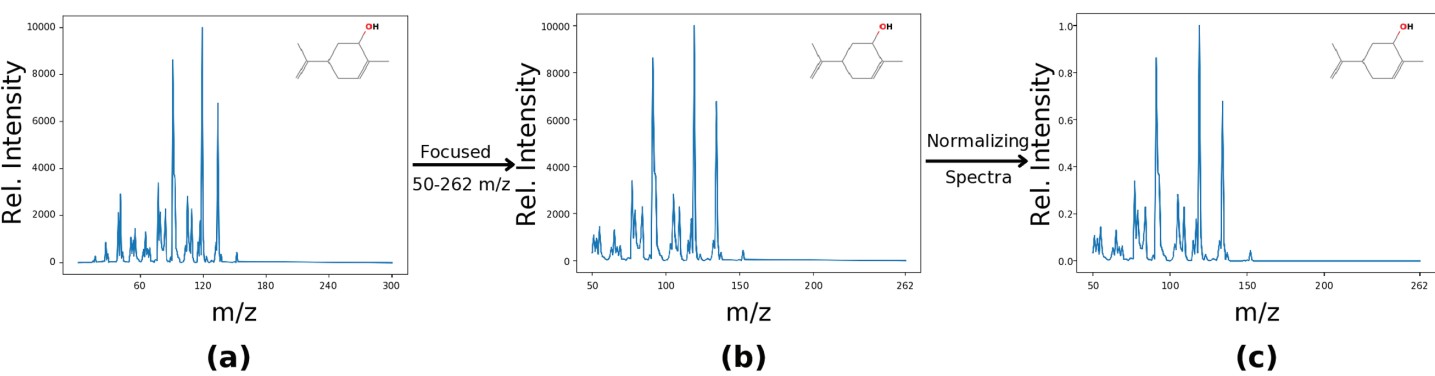

**Fig 3. Overview of feature generation from MS for molecule Carveol (shown in inset).** **(a)** Original MS from NIST **(b)** Focused intensity range of 50–262 m/z, yielding 213 intensities. **(c)** Normalization of the Mass Spectra.

## 2.3 Classification models

According to the histogram in Fig 1, the most frequent label is "fruity", while the least frequent is "fennel/cedar", This indicates that the dataset has a notable class imbalance. Instead of using random sampling, we used iterative stratified sampling to guarantee a balanced representation of the data in both the training and test sets. This approach, implemented using the 'iterative_train_test_split' function from the scikit-multilearn library [28], helps to prevent the unintentional exclusion of minority odor descriptors that can occur with random sampling, which may worsen label imbalance. Figures 4(a) and 4(b) illustrates splitting of data into training and test sets done using iterative and random sampling methods, respectively. In the figure the line charts compare these two types of splits, showing that the non-overlapping orange and blue lines represent differences in label occurrences between the training and test sets. The results demonstrate that the iterative stratified sampling provides a more representative split of the training and test datasets.

**Cost-sensitive multilayer perceptron model (CSMLP).** We used a cost-sensitive multi-layer perceptron (CSMLP) for our study because of the notable class imbalance in our dataset. It is assumed that the costs of misclassification (false positives and false negatives) are equivalent for conventional multilayer perceptrons (MLPs) trained using the backpropagation of error algorithm. To overcome this difficulty, we used the CSMLP since a false negative is more expensive than a false positive [22]. To better account for class imbalance, we adjusted the loss function of the MLP in our suggested CSMLP. In particular, we increased the penalty for

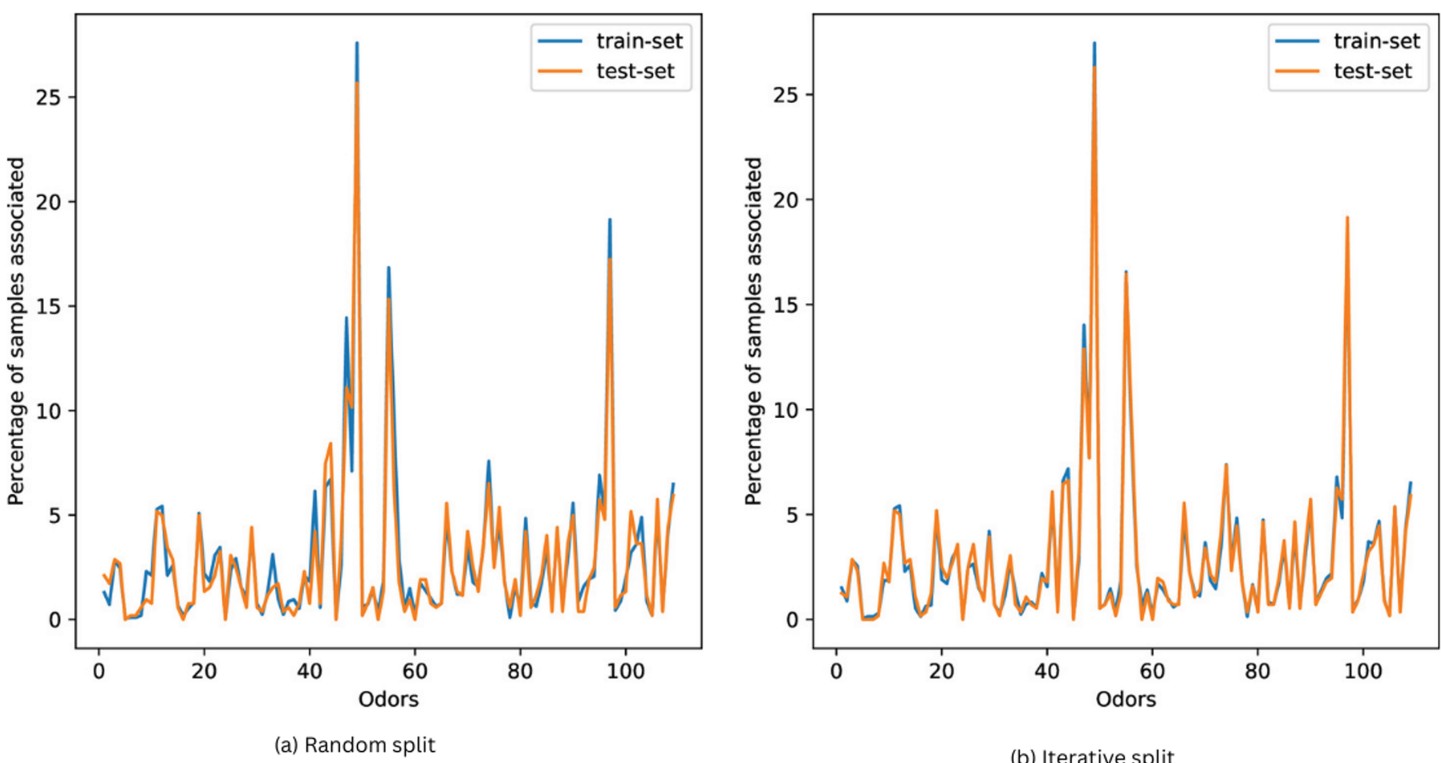

(a) Random split  (b) Iterative split

**Fig 4.** Subset-IGD dataset split into training and test sets using random and iterative sampling. The number on the x-axis signifies the order of odor labels same as in Fig 1.

errors on minority-class samples by using focal loss [29]. Each sample is given a weight by focal loss according to its difficulty, which is determined by the loss that the CSMLP experiences on that sample. Higher loss samples are seen as more difficult. The following is the focal loss (FL) equation:

$$FL = -\frac{1}{N} \sum_{i=1}^{N} \sum_{j=1}^{C} \Bigg( (1 - y_{\text{pred,ij}})^{\gamma} \cdot y_{\text{true,ij}} \cdot \log(y_{\text{pred,ij}} + \epsilon)$$

$$+ (y_{\text{pred,ij}})^{\gamma} \cdot (1 - y_{\text{true,ij}}) \cdot \log(1 - y_{\text{pred,ij}} + \epsilon) \Bigg) \tag{2}$$

The parameter $\gamma$ serves as the focusing parameter in Focal Loss. **C** represents the number of classes, while **N** denotes the number of samples in a batch. The true label for the jth class of the ith example is represented by the variable $y_{\text{true,ij}}$, and the predicted probability for the jth class of the ith example is represented by the variable $y_{\text{pred,ij}}$. A tiny constant called $\epsilon$ is also employed to avoid numerical instability.

We used a fully connected layer CSMLP model. We added a sigmoid function at the last layer and used ReLU for the activation function. Dropout was used as a regularization strategy to assist avoid overfitting, and the models were trained using the Adam optimizer (cited in Keras). The detailed architecture of the model is provided in Supplementary File 2.

**Random forest.** Several decision trees comprise the ensemble classification model, Random Forest (RF) [19]. It is created via random variable selection and bagging. The principle behind constructing trees in Random Forest is similar to that of decision trees, relying on recursive partitioning. In recursive partitioning, the distribution of observations in the training sample is crucial for determining the precise location of the cut-point and selecting the splitting variable. Since small changes in the training data can influence the choice of the initial split variable or cut-point, decision trees tend to be unstable classification models. Consequently, the structure of a decision tree will change as well. Random Forest addresses this instability by combining multiple trees. While a single decision tree may be affected by fluctuations, averaging the predictions of numerous trees leads to more reliable results. However, if the trees in the forest are too similar, the model's classification accuracy can decline. Therefore, diversifying the tree population is essential for enhancing the model's categorization performance. Random Forest achieves this diversification through the randomization of both the input variable sets and the training data sets. RF is commonly used as a baseline model for performance comparison in studies that involve multi-label datasets [8]. Details regarding the model's hyperparameters can be found in supplementary file 2.

**Binary relevance.** The Binary Relevance (BR) approach breaks down a multi-label classification problem into a sequence of binary classification problems, one for each label $y_j$ in the dataset. In particular, a distinct binary classification issue is generated for every label $y_j$. If $y_j$ appears in the matching multi-label example, the label for each example in the training set is regarded as positive; if not, it is seen as negative. By combining the classification results from all the binary classifiers, we can determine the final labels for each multi-label example.

**Classifier chains model.** As in BR, the Classifier Chains (CC) model uses L binary transformations [21], one for each label. In this respect, CC is similar to BR in that it forms a classifier chain by extending the attribute space of each binary model with the 0/1 label relevances of all prior classifiers. CC is different from the BR as it can exploit correlations among labels.

## 2.4 Measures to assess data imbalance

This section discusses the unique traits of imbalanced multi-label datasets and describes the suggested metrics for determining their degree of imbalance.

**Imbalance characteristics in multi-label datasets.** Each data point in a multi-label dataset is often linked to several labels. The likelihood that some labels are uncommon or frequent increases with the number of labels. Fig 1 illustrates this for Subset-IGD and IGD datasets. The figure's bars display the distribution of labels, ranging from the most common (left) to the least common (right). Nonetheless, inferring the imbalance level from metrics like Card and Dens—the most frequently used in the literature to describe multi-label datasets— is not straightforward. Each sample of this multi-label dataset is related with somewhat more than three labels on average, according to Table 2's 3.102 Card value of IGD. Therefore, in a single occurrence, any of its 109 labels may appear with two or three other labels. Although neither Card nor Dens are designed to take this into account, this includes pairings in which the labels occur most and least frequently together. A graphical depiction alone is not a reliable way to assess the degree of imbalance [30]. As a result, precise measurements are required to determine the degree of imbalance in multi-label datasets.

Regarding binary classification, we typically measure imbalance based on just two classes: the majority and minority classes. However, multi-label datasets often contain hundreds of labels, some with very low or high frequencies. Hence, it's crucial to assess the imbalance in multi-label classification (MLC) across all labels, not just two. To address this, we use the following metrics, as detailed by Charte et al. [30].

**IRLbl: Imbalance ratio per label.** According to Eq (3), the IRLbl value for label y in a multi-label dataset (D) with labels Y and $Y_i$ as the *i-th* label is the ratio between the majority label and the label y. For the most frequent label, this ratio is 1, while for others, it is higher. Greater label imbalance is indicated by higher IRLbl values [30].

$$IRLbl(y) = \frac{argmax_{y'=Y_1}^{Y_{|Y|}} \left( \sum_{i=1}^{|D|} h(y', Y_i) \right)}{\sum_{i=1}^{|D|} h(y, Y_i)}, \quad h(y, Y_i) = \begin{cases} 1 & \text{if } y \in Y_i \\ 0 & \text{if } y \notin Y_i. \end{cases} \quad (3)$$

**MeanIR: Mean imbalance ratio.** The measure gives an average imbalance level in a multi-label dataset, as in Eq (4). Different distributions in label can lead to the same MeanIR value [30]. Therefore, it should be used alongside other measures. Expression for MeanIR is given below:

$$\text{MeanIR} = \frac{1}{|Y|} \sum_{y=Y_1}^{Y_{|Y|}} (IRLbl(y)). \quad (4)$$

**CVIR: Coefficient of variation of IRLbl.** This coefficient reflects the variation in IRLbl, as described in Eq (5). It is used to evaluate whether all labels are subject to similar levels of imbalance or if there are notable differences among them. A higher CVIR value indicates greater disparities in imbalance among the labels [30].

$$\text{CVIR} = \frac{\text{IRLbl}_\sigma}{\text{MeanIR}}, \quad \text{IRLbl}_\sigma = \sqrt{\frac{\sum_{y=Y_1}^{Y_{|Y|}} (IRLbl(y) - \text{MeanIR})^2}{|Y| - 1}} \quad (5)$$

## 2.5 Resampling techniques

Resampling methods are often used to address imbalanced data in multi-label datasets (MLDs). These methods involve preprocessing the MLDs by either removing samples from

the majority label (undersampling) or creating new samples for the minority label (oversampling) [30]. Sometimes, a combination of both methods is used. These methods can be categorized as random or heuristic, depending on how samples are added or removed. Random methods choose samples to delete or create randomly, while heuristic methods use specific rules to identify and create the right instances. In this study, we specifically utilize random resampling methods. Random resampling methods for multi-label classification (MLC) can be based on techniques such as the Label Powerset (LP) transformation, Binary Relevance (BR) methods, and various imbalance measures. However, LP-based resampling may be limited due to the sparsity of labels in multi-label datasets (MLDs). In these datasets, there can be as many unique label combinations as there are instances, leading to scenarios where all label sets can simultaneously be classified as both majority and minority cases. An alternative approach to address this challenge is focusing on each individual label's imbalance level. Examples of such methods include Multi-Label Random Under-Sampling (ML-RUS) and Multi-Label Random Over-Sampling (ML-ROS), which concentrate on the frequency of individual labels rather than on entire label sets. These methods specifically target instances that contain one or more minority labels. In this study, we utilize ML-RUS and ML-ROS [30]. To determine the imbalance level of each label, we use metrics such as IRLbl, MeanIR, and CVIR. Labels with IRLbl values higher than MeanIR are referred to as minority labels, indicating which instances should be cloned or not removed. Labels with IRLbl values lower than MeanIR are identified as majority labels.

**ML-ROS: Individual label random oversampling.** ML-ROS (see Algorithm 1) uses the IRLbl metric to find minority labels (i.e., IRLbl > MeanIR). It randomly selects instances from these minority labels to clone. The algorithm constantly updates IRLbl during each cycle, so if a minority label crosses MeanIR, it stops being cloned. It's crucial to remember that non-minority labels could also be present in cloned instances, which could increase the frequency of these labels in the dataset.

**ML-RUS: Individual label random undersampling.** ML-RUS also uses the IRLbl metric to find minority labels and protect their samples from being removed. It randomly selects instances for deletion. Like ML-ROS, ML-RUS updates the IRLbl of affected labels after each deletion, except for those that have reached the MeanIR threshold.

## 2.6 Feature importance analysis

Explainable AI (XAI) methods are meant to explain and interpret the black box ML/DL models. We use SHAP (SHapley Additive exPlanations) as a technique for feature importance analysis [31]. SHAP is a model-agnostic method and does not require model retraining [32]. SHAP excels with data like time series and 1D datasets due to its ability to capture temporal dependencies and sequential patterns.

**SHapley Additive exPlanations (SHAP).** SHAP as mentioned in Data Analysis section is used for saliency analysis of the model's prediction. SHAP's approach to feature importance offers a fresh perspective compared to traditional methods like permutation and Gini index [33]. While traditional methods focus on the impact of feature permutations on model performance, SHAP evaluates the contribution of each feature directly to predictions. This nuanced distinction enhances our understanding of feature importance, especially in complex models. Additionally, traditional feature importance plots often lack depth, providing limited insight beyond ranking features. In contrast, SHAP's global explanations offer more reliable results, making it a promising alternative for feature importance in various contexts.

In essence, the Shapley value named after the economist who first proposed is a technique of allocating profits to stakeholders fairly, taking into account their contributions [34]. It has

**Algorithm 1.** Pseudo-code for ML-ROS Algorithm

**Inputs:** ⟨Dataset⟩$D$, ⟨Percentage⟩$P$
**Outputs:** Preprocessed dataset
1: $samplesToClone \leftarrow |D|/100 * P$                    ▷ P% size increment
2: $L \leftarrow \texttt{labelsInDataset}(D)$          ▷ Obtain the full set of labels
3: $MeanIR \leftarrow \texttt{calculateMeanIR}(D, L)$
4: **for** each label label in L **do**          ▷ Sets of minority label samples
5:          $IRLbl_{label} \leftarrow \texttt{calculateIRperLabel}(D, label)$
6:          **if** $IRLbl_{label} > MeanIR$ **then**
7:                    $minSet_{i++} \leftarrow \texttt{Set}_{label}$
8:          **end if**
9: **end for**
10: **while** samplesToClone>0 **do**          ▷ Instances cloning loop
11:      **for** each $minSet_i$ in minSet **do**
12:          $x \leftarrow \texttt{random}(1, |\texttt{minSet}_i|)$
13:          $\texttt{cloneSample}(\texttt{minSet}_i, x)$   ▷ From each minority set, take a random sample and clone it
14:          **if** $IRLbl_{minSet_i} \leq MeanIR$ **then**
15:              $minSet \rightarrow minSet_i$                    ▷ Exclude from cloning
16:          **end if**
17:          $samplesToClone \leftarrow samplesToClone - 1$
18:      **end for**
19: **end while**

the following definition:

$$\phi(x_i) = \sum_{S \subseteq \{1,2,...,K\} \setminus \{i\}} \frac{|S|! \cdot (K - |S| - 1)!}{K!} [f_x(S \cup \{i\}) - f_x(S)] \tag{6}$$

in which the number of stakeholders is $K$. The difference between the profit made by group $S$ members alone ($f_x(S)$) and the profit made by entity $i$ and the group members ($f_x(S \cup \{i\})$) is the difference that represents the marginal contribution of entity $i$, as indicated by the bracketed portion of Eq 6. The average of the marginal contributions from each combination is used to get the Shapley value, and this process is repeated for every possible combination.

The Shapley value is sole profit allocation strategy that satisfies all four requirements—linearity, symmetry, efficiency and null player. The concept of Shapley value is utilized and SHapley Additive exPlanation (SHAP) are derived. SHAP represents the outcome of patient $j$, denoted as $f(x^{(j)})$, as the total contributions from each feature $i$, denoted as $\phi_i(x_i^{(j)})$.

$$\phi_0 = E[f(x^{(j)})] = \frac{1}{N} \sum_{j=1}^{N} f(x^{(j)}) \tag{7}$$

$$\phi_i(x_i^{(j)}) = \phi(x_i^{(j)}) - \frac{1}{N} \sum_{k=1}^{N} \phi(x_i^{(k)}) \tag{8}$$

$$f(x^{(j)}) = \phi_0 + \sum_{i=1}^{K} \phi_i(x_i^{(j)}) \tag{9}$$

where $N$ is the number of patients. It can be derived that for all $i$, $E(\phi(X_i)) = \frac{1}{N} \sum_{j=1}^{N} \phi(x_i^{(j)}) = 0$ from Eq 8.

The SHAP value has been proven to be reliable and suitable for all machine learning techniques, including GLM (Generalized Linear Models) [34]. Nevertheless, when the number of characteristics $K$ increases, the computation time of naive SHAP calculations grows exponentially.

In summary, SHAP offers three primary advantages over traditional methods for feature importance [35]:

1. SHAP aggregates values to demonstrate the individual impact of each predictor, whether positive or negative, providing global interpretability.
2. Each data point is associated with its unique set of SHAP values, enhancing transparency by offering insights into local interpretability. In contrast, traditional techniques often provide insights only at the population level.
3. SHAP's efficient implementation allows for fast computation across various tree-based models. Conversely, other methods may require alternative models such as logistic regression or linear regression for feature importance determination.

We utilized SHAP feature importance scores to analyze predictions made by the Cost-Sensitive Multilayer Perceptron Model (CSMLP), which was trained on VS, MS and FP features. The identified important features were visually represented by overlaying them on the spectra. Additionally, we created SHAP Summary plots and Bar Plots to showcase the feature importance of the top three classes with the highest occurrence. ***shap*** python library was used to derive all these results [27].

## 3 Results and discussion

In this section we present results of resampling and classification, followed by results of feature fusion and feature significance analysis.

### 3.1 Classification results.

We trained four distinct classification models, as outlined in the previous sections. These models were trained on the IGD_FP, Subset-IGD_FP, Subset-IGD_VS and Subset-IGD_MS. We used an 80%-20% split for our dataset in all the experiments. The performance of the model was assessed using micro-averaged F1 scores, macro-averaged F1 scores, and the area under the ROC curve (AU-ROC). The results for the macro-averaged F1 scores and AUC-ROC scores are presented in Supplementary File 2. To enhance the model and get the best results, we ran a number of tests and adjusted the hyperparameters. After 100 epochs of end-to-end training, the CSMLP model's weights were kept from the epoch with the greatest F1 score.

We observed significant label imbalance in both the IGD and Subset-IGD datasets. To mitigate this, we used two resampling techniques: ML-RUS and ML-ROS. Table 1 represents the classification results obtained on the baseline dataset and after resampling it. After resampling the datasets at varying percentages, we noted a considerable increase in the F1 scores

**Table 1. This table compares the performance of RF, BR, CC, and CSMLP algorithms across various datasets and features. The evaluation uses Multi-Label Random Under-Sampling (MLRUS) and Multi-Label Random Over-Sampling (MLROS) techniques with sampling ratios of 10%, 20%, and 30%. Micro-averaged F1-scores, Precision and Recall are reported for each algorithm-dataset pair; each cell has F1 score on the top, then Precision in the middle and Recall at the bottom. The last column shows percentage increases from the baseline compared to our best result to provide insights into handling class imbalance and improving classification accuracy. Additionally, the RF, BR, and CC results are compared with the findings from Saini et al. [8] on the IGD_FP dataset.**

| Algorithm | Dataset | Baseline | Multi-Label Random Under-Sampling (MLRUS) | | | Multi-Label Random Over-Sampling (MLROS) | | | % F1 Increase from Baseline |
|---|---|---|---|---|---|---|---|---|---|
| | | | 10 | 20 | 30 | 10 | 20 | 30 | |
| **Random Forest (RF)** | IGD_FP | 0.320 [8]<br>0.375 [8]<br>0.279 [8] | 0.301<br>0.360<br>0.259 | 0.311<br>0.379<br>0.263 | 0.294<br>0.369<br>0.244 | 0.320<br>0.373<br>0.280 | 0.323<br>0.376<br>0.283 | **0.324**<br>0.376<br>0.284 | 1.250 |
| | Subset-IGD_FP | 0.296<br>0.349<br>0.256 | **0.333**<br>0.376<br>0.291 | 0.296<br>0.383<br>0.241 | 0.298<br>0.387<br>0.247 | 0.300<br>0.354<br>0.260 | 0.296<br>0.350<br>0.257 | 0.297<br>0.352<br>0.256 | 12.500 |
| | Subset-IGD_VS | 0.183<br>0.271<br>0.140 | 0.188<br>0.319<br>0.133 | **0.201**<br>0.312<br>0.148 | 0.192<br>0.292<br>0.153 | 0.189<br>0.278<br>0.143 | 0.184<br>0.271<br>0.139 | 0.185<br>0.272<br>0.140 | 9.836 |
| | Subset-IGD_MS | 0.201<br>0.303<br>0.151 | 0.197<br>0.323<br>0.142 | 0.167<br>0.265<br>0.122 | 0.193<br>0.337<br>0.136 | **0.212**<br>0.295<br>0.165 | 0.196<br>0.297<br>0.146 | 0.202<br>0.304<br>0.151 | 5.472 |
| **Binary Relevance (BR)** | IGD_FP | 0.352 [8]<br>0.357[8]<br>0.348[8] | 0.328<br>0.344<br>0.313 | 0.345<br>0.370<br>0.324 | 0.314<br>0.354<br>0.283 | 0.353<br>0.353<br>0.348 | **0.354**<br>0.358<br>0.349 | 0.352<br>0.357<br>0.347 | 1.130 |
| | Subset-IGD_FP | 0.324<br>0.338<br>0.311 | **0.340**<br>0.360<br>0.3236 | 0.317<br>0.365<br>0.280 | 0.334<br>0.373<br>0.303 | 0.320<br>0.333<br>0.308 | 0.323<br>0.335<br>0.312 | 0.323<br>0.338<br>0.309 | 4.938 |
| | Subset-IGD_VS | 0.215<br>0.277<br>0.176 | 0.234<br>0.329<br>0.181 | **0.243**<br>0.323<br>0.161 | 0.195<br>0.313<br>0.142 | 0.212<br>0.279<br>0.171 | 0.212<br>0.277<br>0.171 | 0.211<br>0.277<br>0.170 | 13.023 |
| | Subset-IGD_MS | 0.230<br>0.310<br>0.182 | 0.219<br>0.321<br>0.166 | 0.187<br>0.268<br>0.143 | 0.217<br>0.338<br>0.159 | **0.235**<br>0.312<br>0.176 | 0.229<br>0.310<br>0.181 | 0.228<br>0.310<br>0.180 | 2.173 |
| **Classifier Chain (CC)** | IGD_FP | 0.321 [8]<br>0.376[8]<br>0.292[8] | 0.312<br>0.369<br>0.271 | 0.326<br>0.390<br>0.280 | 0.300<br>0.368<br>0.253 | **0.327**<br>0.374<br>0.290 | 0.326<br>0.373<br>0.291 | 0.325<br>0.372<br>0.292 | 1.869 |
| | Subset-IGD_FP | 0.318<br>0.368<br>0.280 | **0.340**<br>0.401<br>0.295 | 0.299<br>0.385<br>0.245 | 0.312<br>0.386<br>0.262 | 0.321<br>0.373<br>0.282 | 0.315<br>0.368<br>0.276 | 0.317<br>0.371<br>0.277 | 6.918 |
| | Subset-IGD_VS | 0.205<br>0.293<br>0.158 | 0.213<br>0.339<br>0.155 | **0.229**<br>0.3294<br>0.270 | 0.216<br>0.319<br>0.257 | 0.201<br>0.292<br>0.157 | 0.202<br>0.290<br>0.155 | 0.203<br>0.291<br>0.156 | 11.165 |
| | Subset-IGD_MS | 0.221<br>0.320<br>0.169 | 0.216<br>0.332<br>0.160 | 0.176<br>0.271<br>0.130 | 0.215<br>0.348<br>0.156 | **0.223**<br>0.322<br>0.170 | 0.220<br>0.320<br>0.168 | 0.214<br>0.313<br>0.163 | 0.904 |
| **Cost Sensitive MLP (CSMLP)** | IGD_FP | 0.401<br>0.355<br>0.477 | 0.395<br>0.376<br>0.415 | 0.386<br>0.344<br>0.440 | 0.372<br>0.329<br>0.428 | **0.410**<br>0.354<br>0.488 | 0.409<br>0.373<br>0.456 | 0.402<br>0.350<br>0.472 | 2.244 |
| | Subset-IGD_FP | 0.392<br>0.391<br>0.392 | **0.400**<br>0.352<br>0.463 | 0.387<br>0.326<br>0.474 | 0.377<br>0.333<br>0.436 | 0.390<br>0.354<br>0.433 | 0.395<br>0.324<br>0.506 | 0.394<br>0.354<br>0.443 | 2.040 |
| | Subset-IGD_VS | 0.305<br>0.252<br>0.387 | **0.346**<br>0.313<br>0.386 | 0.317<br>0.260<br>0.407 | 0.329<br>0.289<br>0.382 | 0.310<br>0.258<br>0.386 | 0.310<br>0.254<br>0.396 | 0.304<br>0.235<br>0.430 | 13.442 |
| | Subset-IGD_MS | 0.297<br>0.265<br>0.337 | 0.283<br>0.250<br>0.327 | 0.274<br>0.233<br>0.332 | 0.277<br>0.255<br>0.302 | 0.299<br>0.263<br>0.347 | 0.295<br>0.271<br>0.323 | **0.301**<br>0.278<br>0.329 | 1.346 |

**Table 2. Table shows values of imbalance measures for Integrated Dataset (IGD) and Subset of Integrated Dataset (Subset-IGD) for original data, and resampled data using ML-RUS (10%) and ML-ROS (10%).**

| Method | IGD | | | | | Subset-IGD | | | | |
|---|---|---|---|---|---|---|---|---|---|---|
| | Card | Dens | CVIR | MaxIR | MeanIR | Card | Dens | CVIR | MaxIR | MeanIR |
| Original data | 3.102 | 0.028 | 1.306 | 227.77 | 29.169 | 3.164 | 0.029 | 2.405 | 709.0 | 49.403 |
| Resampled using ML-RUS | 3.098 | 0.028 | 1.264 | 229.375 | 28.770 | 3.171 | 0.029 | 2.295 | 643 | 47.854 |
| Resampled using ML-ROS | 3.091 | 0.028 | 0.950 | 121.933 | 23.707 | 3.171 | 0.029 | 2.272 | 631 | 31.589 |

**Table 3. Table shows the Feature Fusion Results achieved by concatenating different data modalities, taken in pairs and all three together. It also highlights the percentage increase in F1 score of fused features compared to the best single modality result among the fused features.**

| Dataset | Precision | Recall | F1 Score | Best Single Modality F1 Score | F1 Increase (%) |
|---|---|---|---|---|---|
| Subset-IGD_VS + Subset-IGD_MS | 0.264 | 0.394 | 0.316 | 0.307 (on Subset-IGD_VS) | 2.931 |
| Subset-IGD_VS + Subset-IGD_FP | 0.359 | 0.434 | 0.398 | 0.394 (on Subset-IGD_FP) | 1.015 |
| Subset-IGD_MS + Subset-IGD_FP | 0.365 | 0.441 | 0.400 | 0.394 (on Subset-IGD_FP) | 1.522 |
| Subset-IGD_VS + Subset-IGD_MS + Subset-IGD_FP | 0.350 | 0.457 | 0.399 | 0.394 (on Subset-IGD_FP) | 1.269 |

for all four models; this highlights the effectiveness of resampling techniques in mitigating label imbalances in multi-label datasets. On the IGD_FP dataset, the combination of resampling with RF, BR, and CC performed noticeably better than any of the models used by Saini et al. [8]. The CSMLP performed noticeably better than any other model. Deep learning models like CSMLP can accommodate extremely complicated data that typical machine learning models are unable to capture due to their many adjustable parameters. The CSMLP model trained on IGD_FP resampled with ML-ROS (10%) achieved the highest F1 score of 0.410 on the smell dataset, underscoring the potential of deep learning in olfaction. Additionally, the CSMLP algorithm with resampling outperformed all models trained on different modalities. We also observed a comparative performance of the models on the Subset-IGD_VS and Subset-IGD_MS datasets, Subset-IGD_VS performed better than Subset-IGD_MS, highlighting the crucial role of spectral data in explaining the olfaction mechanism. Table 2 shows the effect of resampling on various imbalance measures, for brevity we show scores for datasets resampled with ML-ROS (10%, 20%, 30%) and ML-RUS (10%, 20%, 30%) as they gave the highest F1 scores and compared their scores with baseline model without any sampling. Table 2 shows that both CVIR and MeanIR have significantly reduced after the resampling on both the dataset with ML-ROS and ML-RUS.

### 3.2 Feature fusion results.

We conducted an exploratory analysis focusing on feature fusion, which involved combining three types of features through simple concatenation methods. The resulting predictive performance of these fused features is outlined in Table 3, where the micro-averaged F1-scores are displayed. We used the best-performing model CSMLP with an 80-20 dataset split. Table 3 clearly demonstrates that concatenating different modalities captures important features, leading to noticeable improvements in the results. Also, in future studies, a more advanced feature fusion strategy can be utilized for better performance.

### 3.3 Feature importance analysis.

SHAP (Shapley Additive Explanations) was used for determining feature importance in CSMLP for Subset-IGD_VS, Subset-IGD_MS and Subset-IGD_FP. Fig 5(a), Fig 6(a), Fig 7(a) presents the SHAP summary plots for the "Fruity" class, across three distinct data modalities:

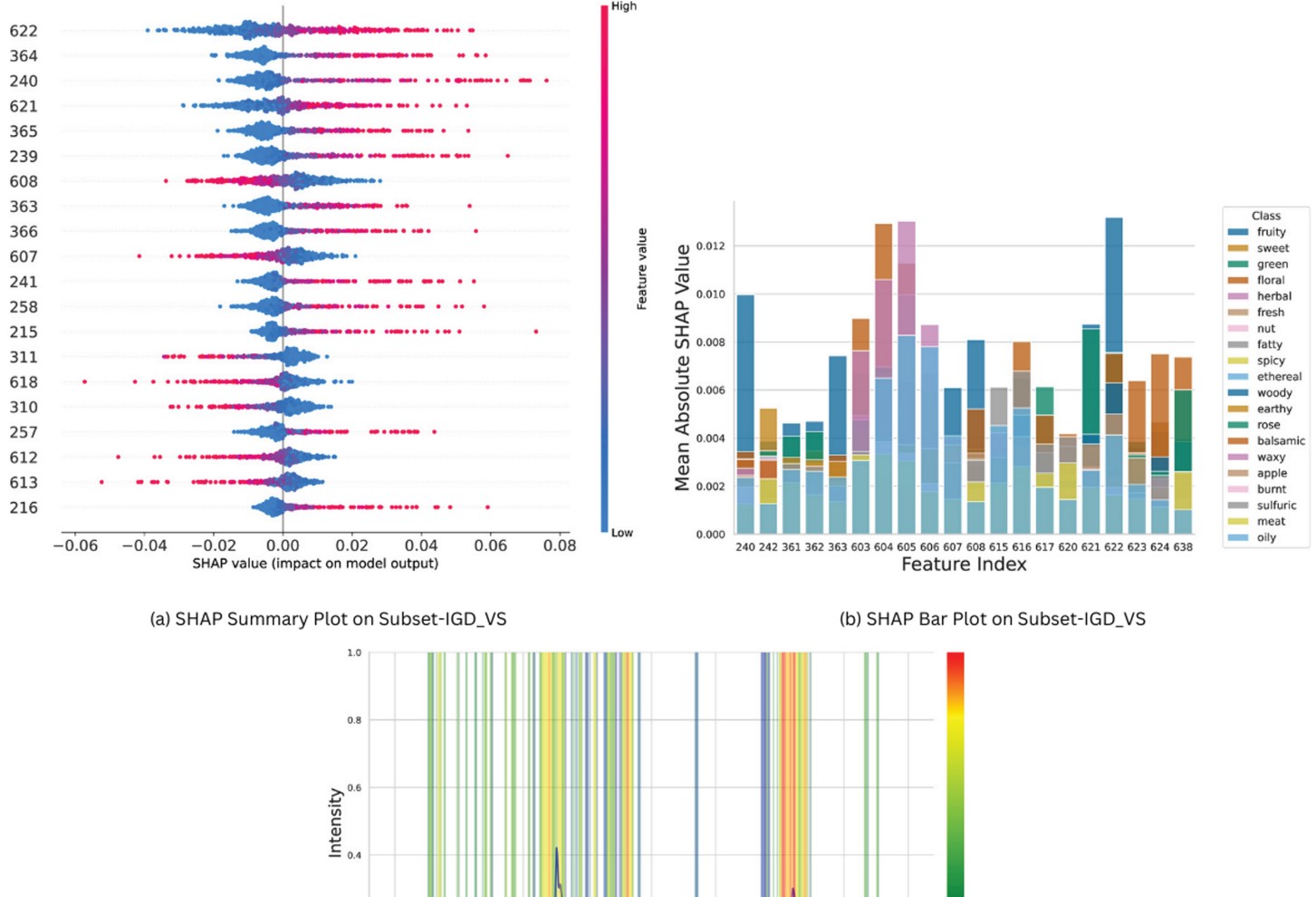

(a) SHAP Summary Plot on Subset-IGD_VS

(b) SHAP Bar Plot on Subset-IGD_VS

(c) SHAP Feature Importance Plot on Subset-IGD_VS

**Fig 5. Feature Importance plots for Subset-IGD_VS: (a)** Summary plot between SHAP feature importance values v/s 20 most important features illustrating SHAP feature importance values for Subset-IGD_VS data modality on CSMLP model, focusing on "Fruity" odor class. **(b)** SHAP Bar plot between Feature Index v/s Mean Absolute SHAP Value showcasing top 20 significant features from Subset-IGD_VS across 20 predominant classes in the dataset. **(c)** Frequency v/s Intensity Overlay of SHAP-extracted best features onto averaged Subset-IGD_VS spectra of 5 molecules from predominant classes for assessment of validity and significance.

Subset-IGD_VS, Subset-IGD_MS, and Subset-IGD_FP respectively. We chose the "Fruity" smell as it is most frequent in our dataset, constituting the largest portion of labels in the datasets (comprising more than 27% of molecules). These summary plots provide information on how different features relate to the model's predictions for the "Fruity" odor category. The hyper-parameters used for SHAP analysis are represented in Table 4. Notably, features with negative values tend to exert a negative influence on the model's prediction, while those with

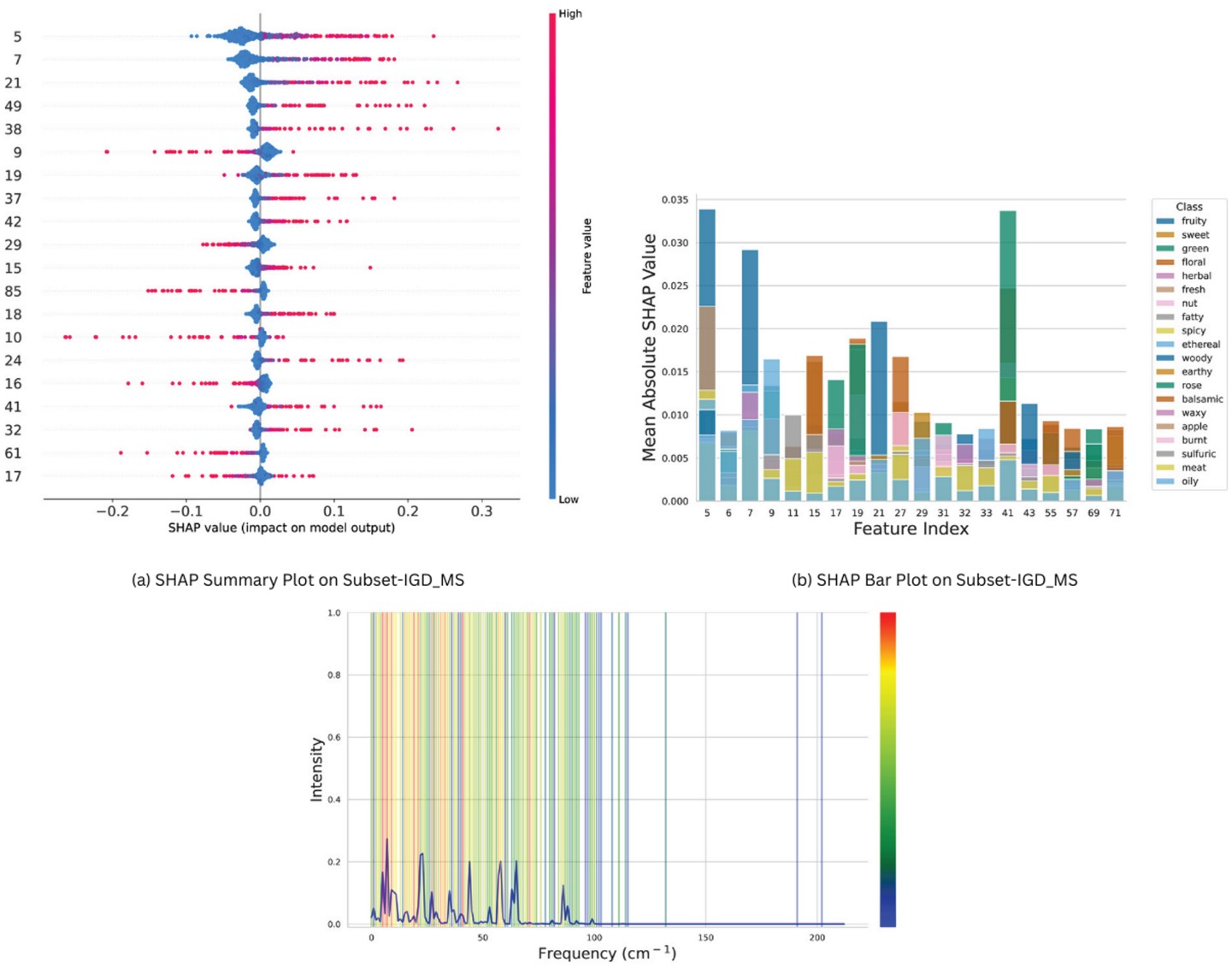

**Fig 6. Feature Importance plots for Subset-IGD_MS: (a)** Summary plot between SHAP feature importance values v/s 20 most important features illustrating SHAP feature importance values for Subset-IGD_MS data modality on CSMLP model, focusing on "Fruity" odor class. **(b)** SHAP Bar plot between Feature Index v/s Mean Absolute SHAP Value showcasing top 20 significant features from Subset-IGD_MS across 20 predominant classes in the dataset. **(c)** Frequency v/s Intensity Overlay of SHAP-extracted best features onto averaged Subset-IGD_MS spectra of 5 molecules from predominant classes for assessment of validity and significance.

positive values contribute positively towards the model prediction. This nuanced analysis provides a deeper understanding of the feature importance dynamics within the "Fruity" class. For brevity, we only present XAI results for "Fruity" here, for "Sweet" and "Green" categories refer to Supplementary file 2.

The Mean Average SHAP Value is considered as the Feature importance score in SHAP analysis. After performing shap analysis we selected the top 20 important features from each odor class, hence we got 230, 102, and 366 unique features for Subset-IGD_VS, Subset-IGD_MS and Subset-IGD_FP respectively, and they were assigned weights proportionate to

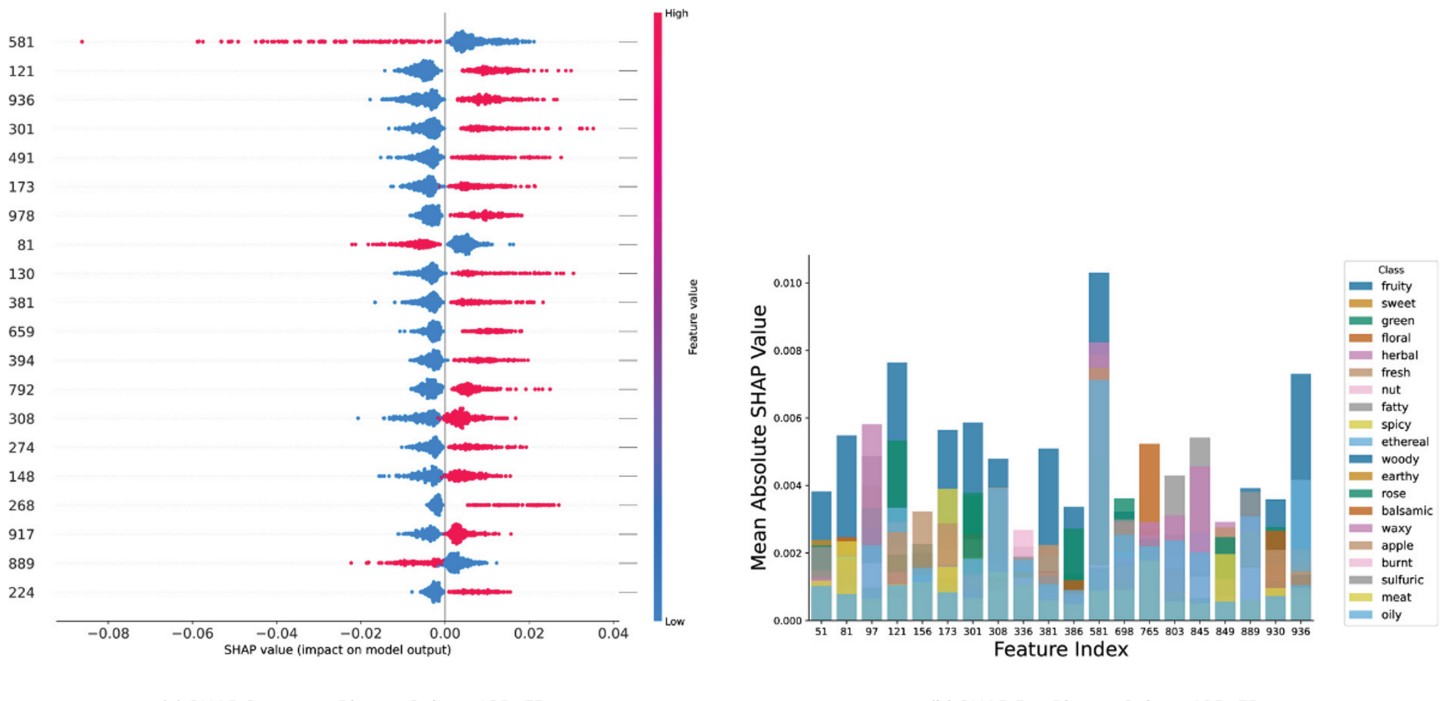

(a) SHAP Summary Plot on Subset-IGD_FP

(b) SHAP Bar Plot on Subset-IGD_FP

**Fig 7. Visual Analysis of Feature Importance in Subset-IGD_FP Dataset: (a)** Summary plot between SHAP feature importance values v/s 20 most important features illustrating SHAP feature importance values for Subset-IGD_FP data modality on CSMLP model, focusing on "Fruity" odor class. **(b)** SHAP Bar plot between Feature Index v/s Mean Absolute SHAP Value showcasing top 20 significant features from Subset-IGD_FP across 20 predominant classes in the dataset.

their frequency. These weights were used in plotting bar plots and plots of overlayed spectra. Bar plots Fig 5(b), Fig 6(b), Fig 7(b) presents the relationship between the Mean Average SHAP values of the top 20 most prevalent odor classes and the top 20 important features that were obtained using the above-mentioned technique across the three data modalities. This visualization serves as a comprehensive lens through which we delve into the model predictions for the important classes and features, providing valuable insights into their interplay and significance.

In Fig 5(c), Fig 6(c), we overlay the important features identified by SHAP, weighted 230 and 102 unique features for Subset-IGD_VS and Subset-IGD_MS onto the averaged spectra of Benzene, 1-ethoxy-4-methyl-, Ethanol, 2-phenoxy-, Fructose, 3-Nonanone, Butanedioic acid, and dimethyl ester. The selection of these molecules is based on the presence of the predominant odor classes in them. Consequently, their averaged spectra provide a comprehensive illustration of how the best features are justified. Features with higher weights were marked red and gradually transitioning to blue gradients as weight decreased. These plots illustrate that the regions of interest for the model coincide with those containing the most significant physical information. This observation underscores the model's capability to accurately identify crucial regions within the spectra, thereby eliminating the concerns of random results or hallucinations.

**Table 4. SHAP Analysis Hyper-Parameters with Descriptions and Impact**

| Parameter | Default Value | Value Used | Alternatives | Description and Impact |
|---|---|---|---|---|
| `model_predict_proba` | N/A | Custom function | `predict`, `predict_proba` | Converts model output into a format suitable for SHAP analysis. Probabilities enhance interpretability. |
| `data` (background data) | N/A (Required) | `Test Data` | Subset of training data | Acts as a reference for SHAP values; a representative sample ensures stable results. |
| `algorithm` | `auto` | `auto` | `exact`, `permutation` | `exact` provides precise values but is slower; `permutation` offers faster approximations. |
| `link` | `identity` | `identity` | `logit` | Transforms output for interpretability; `logit` is also useful for classification models. |
| `max_display` | 20 | 20 | Any integer value | Limits features shown; higher values may clutter the plot while offering more insights. |
| `features` | N/A | `Test Data` | Any 2D data array | Input data for SHAP analysis, must match model features for meaningful results. |
| `feature_names` | N/A | `Test Data Feature Names` | List of feature names | Improves interpretability by using meaningful names instead of indices. |
| `plot_type` | `dot` | `dot` | `bar`, `violin` | `dot` plots show individual feature impacts; `bar` aggregates global importance. |
| `show` | `True` | `False` | `True` | Controls immediate display of the plot; set to `False` for further customization. |
| `shap_values` | N/A | `shap_values[:, :, 96]` | SHAP values for other labels | Extracts values for specific labels in multi-label tasks, with each label having its own set. 96 here represents 96th label out of 109. |
| `dpi` (for figure) | 100 | 300 | Any integer value | Sets resolution of saved plots; higher DPI ensures better clarity for publication. |

## Conclusion

Using machine learning and deep learning models, the study attempts to fill two gaps in the prediction of odorant perception by humans. The first is to compare the predictive performance of three frequently used molecular features (VS, MS, and fingerprint features), and the second is to address the inherent challenge of data imbalance in all the datasets used to predict human perceptions. To this end, we assembled a novel dataset of 2606 molecules and used four different kinds of models to evaluate the performance; results established superior performance of fingerprint features, then VS, and lastly, MS. We used random resampling techniques to address the data imbalance. In order to comprehend the connection between features and odor, we also performed a saliency analysis on the deep learning model. A significant limitation of our study is that it is restricted to only 2,606 molecules; a larger dataset could provide deeper insights into the relationship between odor features. Also, advanced methods could be tried for multi-modal classification and fusion, for we used only baseline models and a simple feature concatenation method. The results of this comprehensive and systematic study have the potential to guide the development of the electric nose.

## Supporting information

**Supplementary File 1. Supplementary File 1 is an Excel spreadsheet containing a list of molecules found in IGD, along with the molecules of Subset-IGD.**
(XLSX)

**Supplementary File 2. Supplementary File 2 is a PDF containing details of model architecture, additional classification results, resampling results on other percentages, SHAP Summary Pots for 2 odors ("Sweet", "Green") across 3 data modalities.**
(PDF)

## Author contributions

**Conceptualization:** Durgesh Ameta, Laxmidhar Behera, Aniruddha Chakraborty, Tushar Sandhan.

**Data curation:** Durgesh Ameta, Surendra Kumar, Rishav Mishra.

**Investigation:** Aniruddha Chakraborty, Tushar Sandhan.

**Methodology:** Durgesh Ameta, Surendra Kumar, Rishav Mishra.

**Project administration:** Laxmidhar Behera.

**Supervision:** Laxmidhar Behera, Aniruddha Chakraborty, Tushar Sandhan.

**Writing – original draft:** Durgesh Ameta, Surendra Kumar, Rishav Mishra.

**Writing – review & editing:** Laxmidhar Behera, Aniruddha Chakraborty, Tushar Sandhan.

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
