## [Decision Letter · Decision Letter 0]

27 Sep 2024

PONE-D-24-27679

Odor Classification: Exploring Feature Performance and Imbalanced Data Learning Techniques

PLOS ONE

Dear Dr. Ameta,

Thank you for submitting your manuscript to PLOS ONE. After careful consideration, we have decided that your manuscript does not meet our criteria for publication and must, therefore, be rejected. As you will see from the reviewers' comments below, all reviewers are mainly concerned with the missing comparison with well-established methods and the lack of clarity in describing the chosen methods. Next to other issues raised, the manuscript needs to be thoroughly revised concerning grammar, word usage, and English in particular. Additionally, the manuscript would be markedly improved by a more constrained presentation and lacks the inclusion of recent publications. Thus, although the presented work is of interest, I think the problems raised by the reviewers can be better addressed in a new version than in a revision of the current manuscript. 

I am sorry that we cannot be more positive on this occasion, but hope that you appreciate the reasons for this decision.

Kind regards,

Florian Ph.S Fischmeister

Academic Editor

PLOS ONE

Reviewers' comments:

Reviewer's Responses to Questions

**Comments to the Author**

1. Is the manuscript technically sound, and do the data support the conclusions?

Reviewer #1: Partly

Reviewer #2: Yes

Reviewer #3: Yes

2. Has the statistical analysis been performed appropriately and rigorously?

Reviewer #1: No

Reviewer #2: Yes

Reviewer #3: Yes

3. Have the authors made all data underlying the findings in their manuscript fully available?

Reviewer #1: No

Reviewer #2: Yes

Reviewer #3: No

4. Is the manuscript presented in an intelligible fashion and written in standard English?

Reviewer #1: Yes

Reviewer #2: No

Reviewer #3: No

5. Review Comments to the Author

Reviewer #1: This paper attempts to predict human perception of odors by using classification methods and resampling techniques. The authors evaluated the performance of a self-built dataset (with three features: mass spectra, vibrational spectra, and molecular fingerprint features) using four multi-label classification models. Then, the authors used random resampling techniques and Shapley Additive Explanations (SHAP) analysis to solve the class imbalance problem in multi-label data sets for odor classification.

Some issues that need to be improved are listed below:

1. In this article, the authors did not conduct comparative experiments with well-known methods/models on data classification and resampling techniques. There are already many commonly used methods on data classification and resampling techniques in recent years. So, in your numerical experiments, the comparison of selected algorithms with other state-of-art algorithms must be provided.

2. On page 6, there is an obvious error in formula (2).

3. In Section 2.5, the details of ML-ROS and ML-RUS techniques have to be provided in more details. For example, the descriptions of the steps in Algorithm 1 (ML-ROS Algorithm) are not clear because some variables and technical terms are not defined first. In addition, you are requested to add the algorithmic steps for ML-RUS Algorithm. Moreover, the flowchart of the proposed algorithms should also be given.

4. In the numerical comparisons of Section 3.2, Error rate and Classification Success Index may be added for verification, except Precision, F1 Score and Recall indicators.

5. Some pictures of your numerical results are not clear enough, for example Figure 2, Figure 5 (a)-(c), Figure 6 (a)-(c), and Figure 7 (a)-(b). You should think of better ways to make the pictures of these numerical results clearer and easier to read.

6. In your numerical experiments (Section 3), a full statistical analysis of the numerical results must be presented. Furthermore, it would be better to address the issue of computational complexity and/or computational speed of the studied methods.

7. Most of your references are too outdated, and there is a lack of relevant papers in this field published in journals and/or important conferences in the past three years (from 2022 to 2024). You should be able to easily search the latest relevant works from the Scopus and Web of Science databases.

8. In the Abstract and Introduction section, the contributions of this article must be emphasized in terms of originality, significance, and performance metrics. Moreover, in Section 1, please describe in detail the motivation behind this work.

9. The text layout and format of this manuscript need to be improved. For example, the distinction between many text paragraphs is unclear or confusing, and the titles of some subsections also need to be more relevant to the topic of the corresponding paragraph.

Reviewer #2: In the abstract and conclusion, the contribution of this paper is not well presented. In the conclusion and abstract, highlight the novelty of the paper.

The introduction is weak and should include the research question, the aim of the paper and the contribution.

In related work…. Many researches work on this idea. What is really the novelty as compared to other studies? What is the new and the difference between the previous works and present work?.

Improve the quality of literature along with the latest literature.

The explanation of the related work needs to be criticized and improved in general.

What about last updating in this topic and new references from 2019-2024? The survey of existing literature is not sufficient. It would useful to include in the Introduction of the paper some discussion on other possible real applications of the obtained results.

Figures are not clear. Clear diagrams and figures are required for readers to have clear images.

Improve the quality of figures for better visibility. It is blur that should be adjusted.

Weak conclusion and the future work were Meaningless in this article.

Conclusion should be more specific with improvement writing quality.

A suggestion for future work should be added in the conclusion section.

- Rewrite the references according to journal template.

-Please strictly follow the instructions to the format specified in the journal template for preparing the paper

The format and English writing of this paper should be improved. The paper needs language revision.

Reviewer #3: The work of DURGESH Ameta et al. is focused on the following aspects: the creation of a large and novel dataset containing VS, MS, and physicochemical features; a systematic evaluation of the predictive performance of various features for odor classification; the application of random resampling techniques to address data imbalance within the odor datasets; an exploratory analysis of multi-modal feature fusion for odor classification; and the development of an explainable deep learning model that offers insights into the relationship between features and odor.

Despite the work being valid and well-structured, the reviewer has the following observations:

- why only F1 score has been considered? This classification metric masks insights into specific errors, and ignores true negatives.

- the methods section would benefit from additional insights into the use of the SHAP library: parameters used and related description

- the manuscript needs an English revision.

6. PLOS authors have the option to publish the peer review history of their article (what does this mean?). If published, this will include your full peer review and any attached files.

Reviewer #1: No

Reviewer #2: **Yes: **Bashra Kadihm Oleiwi

Reviewer #3: No

- - - - -

---

## [Author Response · Author response to Decision Letter 1]

16 Nov 2024

Response to all review comments are inlcuded in point-to-point response file. Also included updated Supplementary File 2.

1. Is the manuscript technically sound, and do the data support the conclusions?

The manuscript must describe a technically sound piece of scientific research with data that

supports the conclusions. Experiments must have been conducted rigorously, with appropriate

controls, replication, and sample sizes. The conclusions must be drawn appropriately based on

the data presented.

Reviewer #1: Partly

Reviewer #2: Yes

Reviewer #3: Yes

2. Has the statistical analysis been performed appropriately and rigorously?

Reviewer #1: No

Reviewer #2: Yes

Reviewer #3: Yes

3. Have the authors made all data underlying the findings in their manuscript fully

available?

The PLOS Data policy requires authors to make all data underlying the findings described in

their manuscript fully available without restriction, with rare exception (please refer to the Data

Availability Statement in the manuscript PDF file). The data should be provided as part of the

manuscript or its supporting information, or deposited to a public repository. For example, in

addition to summary statistics, the data points behind means, medians and variance measures

should be available. If there are restrictions on publicly sharing data—e.g. participant privacy or

use of data from a third party—those must be specified.

Reviewer #1: No

Reviewer #2: Yes

Reviewer #3: No

We made available the list of molecules in both datasets; this list can be used to scrape

the spectra’s molecules from the NIST website as referenced in the manuscript.

4. Is the manuscript presented in an intelligible fashion and written in standard English?

PLOS ONE does not copyedit accepted manuscripts, so the language in submitted articles must

be clear, correct, and unambiguous. Any typographical or grammatical errors should be

corrected at revision, so please note any specific errors here.

Reviewer #1: Yes

Reviewer #2: No

Reviewer #3: No

Some issues that need to be improved are listed below:

1. In this article, the authors did not conduct comparative experiments with well-known

methods/models on data classification and resampling techniques. There are already many

commonly used methods on data classification and resampling techniques in recent years. So,

in your numerical experiments, the comparison of selected algorithms with other state-of-art

algorithms must be provided.

We employed several techniques to address the issue of imbalanced datasets, specifically using

resampling and a cost-sensitive MLP using Focal Loss. For resampling, we applied Multi-Label

Random Under-Sampling (ML-RUS) and Multi-Label Random Over-Sampling (ML-ROS). These

techniques have been widely and recently used to tackle imbalances in multi-label data, and to

the best of our knowledge, this is the first time they have been applied in the domain of smell

prediction, positioning them as state-of-the-art methods[1-4].

Our primary goal was to compare the performance across different modalities, which is why we

restricted our analysis to these baseline techniques. We required a resampling approach that

could be consistently applied across all three types of modalities. In addition, we utilized a

modified cost-sensitive MLP to address class imbalance, a method that has been successfully

employed in recent studies.

Given the scope of our current analysis, which already resulted in a 21-page manuscript, we did

not conduct further analysis at this stage. However, following the reviewers’ suggestions, we

plan to incorporate these additional analyses in the future works.

2. On page 6, there is an obvious error in formula (2).

We regret that we had to make several last-minute changes due to the specific format required

by PLOS ONE (which does not allow figures to be included in the main manuscript).

Unfortunately, in our haste, mistakes were made. We are fully prepared to rectify these errors.

3. In Section 2.5, the details of ML-ROS and ML-RUS techniques have to be provided in more

details. For example, the descriptions of the steps in Algorithm 1 (ML-ROS Algorithm) are not

clear because some variables and technical terms are not defined first. In addition, you are

requested to add the algorithmic steps for ML-RUS Algorithm. Moreover, the flowchart of the

proposed algorithms should also be given.

We have cited the relevant papers for the detailed descriptions of the algorithms. However, due

to the length limitations of the manuscript, we provided only a brief overview of the algorithms in

the main text. If length constraints were not a factor, we would be glad to provide the full

algorithmic details.

4. In the numerical comparisons of Section 3.2, Error rate and Classification Success Index may

be added for verification, except Precision, F1 Score and Recall indicators.

Given the nature of the datasets, previous studies on the IGD_FP dataset primarily reported

results using F1 score, precision, and recall, which we considered sufficient. Many published

studies have focused exclusively on these metrics. However, in response to the reviewers'

suggestions, we have included additional performance metrics such as micro-F1, macro-F1,

nDCG score, accuracy, error rate, and both micro and macro AU-ROC in the supplementary

material for the IGD_FP dataset.

5. Some pictures of your numerical results are not clear enough, for example Figure 2, Figure 5

(a)-(c), Figure 6 (a)-(c), and Figure 7 (a)-(b). You should think of better ways to make the

pictures of these numerical results clearer and easier to read.

All of our figures were saved in high resolution; however, as this was my first submission to

PLOS ONE, I was unaware that separately uploading the images could affect their quality. While

submitting next time, we will take extra care to ensure the quality of the figures is maintained.

6. In your numerical experiments (Section 3), a full statistical analysis of the numerical results

must be presented. Furthermore, it would be better to address the issue of computational

complexity and/or computational speed of the studied methods.

We employed a cost-sensitive MLP along with various classification models previously used for

the same dataset, such as Random Forest (RF), Binary Relevance (BR), Classifier Chain (CC),

and Cost-Sensitive MLP (CSMLP). Since time complexity was not a significant concern in this

context, we did not include these results in the main discussion. Additionally, in traditional

models, the time complexity is generally reported in research papers.

7. Most of your references are too outdated, and there is a lack of relevant papers in this field

published in journals and/or important conferences in the past three years (from 2022 to 2024).

You should be able to easily search the latest relevant works from the Scopus and Web of

Science databases.

Digital olfaction is an open problem in an interdisciplinary field, and very few research groups

are actively engaged in this area. We have referenced four recent papers that are directly

related to this field. Notably, the most recent paper published in PLOS ONE on this topic

appeared in 2020. The list of recent related papers we referred in our work are

1. Pandey N, Pal D, Saha D, Ganguly S. Vibration-based biomimetic odor classification.

Scientific Reports. 2021;11(1). doi:10.1038/s41598-021-90592-x.

2. Saini K, Ramanathan V. Predicting odor from molecular structure: a multi-label

classification approach. Scientific Reports. 2022;12(1).

doi:10.1038/s41598-022-18086-y.

3. Debnath T, Nakamoto T. Predicting human odor perception represented by continuous

values from mass spectra of essential oils resembling chemical mixtures. PLOS ONE.

2020;15(6):e0234688. doi:10.1371/journal.pone.0234688.

4. Lee BK, Mayhew EJ, Sanchez-Lengeling B, Wei JN, Qian WW, Little K, et al. A Principal

Odor Map Unifies Diverse Tasks in Human Olfactory Perception.

2022;doi:10.1101/2022.09.01.504602.

5. Liu C, Miyauchi H, Hayashi K. DeepSniffer: A meta-learning-based chemiresistive odor

sensor for recognition and classification of aroma oils. Sensors and Actuators B:

Chemical. 2022;351:130960. doi:10.1016/j.snb.2021.130960.

8. In the Abstract and Introduction section, the contributions of this article must be emphasized

in terms of originality, significance, and performance metrics. Moreover, in Section 1, please

describe in detail the motivation behind this work.

We have revised our paper and enhanced it in accordance with the suggestions provided.

9. The text layout and format of this manuscript need to be improved. For example, the

distinction between many text paragraphs is unclear or confusing, and the titles of some

subsections also need to be more relevant to the topic of the corresponding paragraph.

We have revised our paper and enhanced it in accordance with the suggestions provided.

Reviewer #2: In the abstract and conclusion, the contribution of this paper is not well presented.

In the conclusion and abstract, highlight the novelty of the paper.

The introduction is weak and should include the research question, the aim of the paper and the

contribution.

In related work…. Many researches work on this idea. What is really the novelty as compared to

other studies? What is the new and the difference between the previous works and present

work?.

Although recent studies in the area of odor classification have employed machine learning (ML)

and deep learning (DL) techniques, none have systematically addressed the issue of data

imbalance. Additionally, while there are papers predicting odor perception using various

modalities[5-7], none have compared their predictive performance within a single study.

Consequently, this study is relevant for advancing the field of digital olfaction. Furthermore, the

use of explainable AI enables us to identify the important feature regions for smell prediction.

Improve the quality of literature along with the latest literature.

Digital olfaction is an open problem in an interdisciplinary field with very few research groups

actively engaged in this area. We have referenced four recent papers that are directly related to

this field. Notably, the most recent paper published in PLOS ONE on this topic appeared in

2020. The list of recent related papers we referred in our work are

1. Pandey N, Pal D, Saha D, Ganguly S. Vibration-based biomimetic odor classification.

Scientific Reports. 2021;11(1). doi:10.1038/s41598-021-90592-x.

2. Saini K, Ramanathan V. Predicting odor from molecular structure: a multi-label

classification approach. Scientific Reports. 2022;12(1).

doi:10.1038/s41598-022-18086-y.

3. Debnath T, Nakamoto T. Predicting human odor perception represented by continuous

values from mass spectra of essential oils resembling chemical mixtures. PLOS ONE.

2020;15(6):e0234688. doi:10.1371/journal.pone.0234688.

4. Lee BK, Mayhew EJ, Sanchez-Lengeling B, Wei JN, Qian WW, Little K, et al. A Principal

Odor Map Unifies Diverse Tasks in Human Olfactory Perception.

2022;doi:10.1101/2022.09.01.504602.

5. Liu C, Miyauchi H, Hayashi K. DeepSniffer: A meta-learning-based chemiresistive odor

sensor for recognition and classification of aroma oils. Sensors and Actuators B:

Chemical. 2022;351:130960. doi:10.1016/j.snb.2021.130960.

The explanation of the related work needs to be criticized and improved in general.

We compared our results with the latest paper published on the IGD_FP dataset and we

significantly outperformed their results with the help of resampling and cost-sensitive MLP [6].

The results are reported in Table 1 of main manuscript.

What about last updating in this topic and new references from 2019-2024? The survey of

existing literature is not sufficient. It would useful to include in the Introduction of the paper some

discussion on other possible real applications of the obtained results.

As it is a new and challenging field and very few research group work in the area, still we

referenced four recent papers that are directly related to this field. Even last paper related to this

area published in PlosOne was 2020. The list of papers we refered in our work are

1. Pandey N, Pal D, Saha D, Ganguly S. Vibration-based biomimetic odor classification.

Scientific Reports. 2021;11(1). doi:10.1038/s41598-021-90592-x.

2. Saini K, Ramanathan V. Predicting odor from molecular structure: a multi-label

classification approach. Scientific Reports. 2022;12(1).

doi:10.1038/s41598-022-18086-y.

3. Debnath T, Nakamoto T. Predicting human odor perception represented by continuous

values from mass spectra of essential oils resembling chemical mixtures. PLOS ONE.

2020;15(6):e0234688. doi:10.1371/journal.pone.0234688.

4. Lee BK, Mayhew EJ, Sanchez-Lengeling B, Wei JN, Qian WW, Little K, et al. A Principal

Odor Map Unifies Diverse Tasks in Human Olfactory Perception.

2022;doi:10.1101/2022.09.01.504602.

5. Liu C, Miyauchi H, Hayashi K. DeepSniffer: A meta-learning-based chemiresistive odor

sensor for recognition and classification of aroma oils. Sensors and Actuators B:

Chemical. 2022;351:130960. doi:10.1016/j.snb.2021.130960.

Figures are not clear. Clear diagrams and figures are required for readers to have clear images.

Improve the quality of figures for better visibility. It is blur that should be adjusted.

We apologise for this as it was the first time we are making submission to PlosOne, althoug the

figures are good in quality but as they are needed to be uploaded separately so the quality got

affected.

Weak conclusion and the future work were Meaningless in this article.

Conclusion should be more specific with improvement writing quality.

A suggestion for future work should be added in the conclusion section.

- Rewrite the references according to journal template.

-Please strictly follow the instructions to the format specified in the journal template for preparing

the paper

The format and English writing of this paper should be improved. The paper needs language

revision.

We have revised our paper and enhanced it in accordance with the suggestions provided.

Reviewer #3: The work of DURGESH Ameta et al. is focused on the following aspects: the

creation of a large and novel dataset containing VS, MS, and physicochemical features; a

systematic evaluation of the predictive performance of various features for odor classification;

the application of random resampling techniques to address data imbalance within the odor

datasets; an exploratory analysis of multi-modal feature fusion for odor classification; and the

development of an explainable deep learning model that offers insights into the relationship

between features and odor.

Despite the work being valid and well-structured, the reviewer has the following observations:

- why only F1 score has been considered? This classification metric masks insights into specific

errors, and ignores true negatives.

Usually given the nature of datasets the previously published paper on IGD_FP dataset

presented results using F1 core, Precision, and recall hence we considered them sufficient

enough. There are multiple published studies just including these results, but on reviewers

suggestions we have included micro-f1, macro-f1 and nDGC.

- the methods section would benefit from additional insights into the use of the SHAP library:

parameters used and related description

We have added the hyper-parameters of the SHAP in our revised version.

- the manuscript needs an English revision.

We have revised our paper and enhanced it in accordance with the suggestions provided.

References:-

[1] D. Liang, X. Chen and Y. Zhang, "Diagnosing Electric Vehicle Failures With Long-Tailed Multilabel

Consumer Reviews: A Bilateral-Branch Deep Learning Approach," in IEEE Transactions on Engineering

Management, vol. 71, pp. 14485-14499, 2024, doi: 10.1109/TEM.2024.3421911

[2] A. Umparat and S. Phoomvuthisarn, "Improving Pre-Trained Models for Multi-Label Classification in

Stack Overflow: A Comparison of Imbalanced Data Handling Methods," 2023 20th International Joint

Conference on Computer Science and Software Engineering (JCSSE), Phitsanulok, Thailand, 2023, pp.

464-469, doi: 10.1109/JCSSE58229.2023.1020201

---

## [Decision Letter · Decision Letter 1]

24 Mar 2025

Odor Classification: Exploring Feature Performance and Imbalanced Data Learning Techniques

PONE-D-24-27679R1

Dear Dr. Ameta,

We’re pleased to inform you that your manuscript has been judged scientifically suitable for publication and will be formally accepted for publication once it meets all outstanding technical requirements.

Kind regards,

Upaka Rathnayake, PhD

Academic Editor

PLOS ONE

Additional Editor Comments (optional):

Reviewers' comments:

Reviewer's Responses to Questions

**Comments to the Author**

1. If the authors have adequately addressed your comments raised in a previous round of review and you feel that this manuscript is now acceptable for publication, you may indicate that here to bypass the “Comments to the Author” section, enter your conflict of interest statement in the “Confidential to Editor” section, and submit your "Accept" recommendation.

Reviewer #1: All comments have been addressed

Reviewer #2: All comments have been addressed

2. Is the manuscript technically sound, and do the data support the conclusions?

Reviewer #1: Yes

Reviewer #2: Yes

3. Has the statistical analysis been performed appropriately and rigorously?

Reviewer #1: Yes

Reviewer #2: Yes

4. Have the authors made all data underlying the findings in their manuscript fully available?

Reviewer #1: Yes

Reviewer #2: Yes

5. Is the manuscript presented in an intelligible fashion and written in standard English?

Reviewer #1: Yes

Reviewer #2: Yes

6. Review Comments to the Author

Reviewer #1: The authors have well responded to my previous questions and made significant improvements. Because the quality of this submission has been significantly improved, I suggest that it could be accepted for publication as long as it meets the format requirements of PLOS One.

Reviewer #2: (No Response)

7. PLOS authors have the option to publish the peer review history of their article (what does this mean?). If published, this will include your full peer review and any attached files.

Reviewer #1: No

Reviewer #2: No

---

## [Editor Report · Acceptance letter]

PONE-D-24-27679R1

PLOS ONE

Dear Dr. Ameta,

I'm pleased to inform you that your manuscript has been deemed suitable for publication in PLOS ONE. Congratulations! Your manuscript is now being handed over to our production team.

Kind regards,

on behalf of

Prof. Upaka Rathnayake

Academic Editor

PLOS ONE